# OpenReview forum: "BlueCodeAgent: A Blue Teaming Agent Powered by Automated Red Teaming for CodeGen AI"
_ICML.cc/2026/Conference — ICML 2026 regular_

### Official Review · Reviewer_jo6J · 2026-03-12

**Soundness:** 3
**Presentation:** 3
**Significance:** 3
**Originality:** 3
**Overall Recommendation:** 4
**Confidence:** 3

**Summary:**

The paper presents a plausible and practically motivated agentic defense framework that combines red-teaming-derived knowledge (as constitutions) with dynamic execution checks. The main limitations are around evaluation rigor (possible generation artifacts and limited reporting), reproducibility details, and the practical reliability/cost/security of LLM-generated dynamic tests. With stronger controls and clearer methodological specifics, the work could be a solid contribution to code-LLM security blue teaming.

**Compliance With Llm Reviewing Policy:**

Affirmed.

**Final Justification:**

My final opinion is accept. Because the provided experiments have solved all my concerns.

**Key Questions For Authors:**

See above weakness. I am looking forward to receive more dataset and metric details from author.

**Limitations:**

The main limitation is the unclear explanation about the dataset, and the experiments should be conducted with more valuable metrics.

**Strengths And Weaknesses:**

Strengths
1. Clear end-to-end framing of blue teaming powered by red teaming. The paper proposes a coherent pipeline: diverse red-teaming → knowledge base → retrieval → constitution summarization → blue-team decision, with an additional dynamic-testing tool for code-level vulnerability claims.
2. Diverse threat coverage and multi-task evaluation. Evaluating on four representative risk types (bias instructions, malicious instructions, prompt injection, and code vulnerabilities) strengthens the argument that the approach is general rather than single-task engineered.
3. Constitution mechanism is practically motivated and empirically effective. The case study (bias detection) illustrates why generic “safety reminders” fail and how concrete constitutions can provide better decision boundaries.

Weaknesses
1. Details of dataset construction and sizes are insufficient for reproducibility and for judging difficulty. The paper introduces BlueCodeKnow/BlueCodeEval but (in the provided excerpt) does not clearly report: total counts per subset, class balance, prompt lengths, deduplication, and filtering criteria (e.g., removing near-duplicates across know/eval).
2. The evaluation metric (single F1) may hide safety/usability tradeoffs. Especially for security guardrails, practitioners care about separate TPR/recall on unsafe and FPR on benign. Some tables provide FP/FN for vulnerabilities, but similar breakdowns are missing for the textual tasks where “near-1.0 F1” could still conceal problematic behavior under distribution shift.

---

> ### Author Rebuttal · Authors · 2026-03-30
>
> We sincerely thank you for recognizing the clear end-to-end framing, the diverse threat coverage, the multi-task evaluation, and the practically motivated constitution mechanism. We also appreciate the positive comments on the case study. Below, we address each weakness and question.
>
> > W1&Q: Details of dataset construction and sizes are insufficient for reproducibility
>
> Thank you for the constructive suggestion. We provide the following dataset statistics:
>
> |Subset|#Total|#Unsafe|#Safe|#Cat|Prompt Len (Min/Avg/Max Char)|
> |---|---|---|---|---|---|
> |BlueCodeKnow-Bias|528|276|252|8|43/97/218|
> |BlueCodeEval-Bias|504|252|252|9|37/93/177|
> |BlueCodeKnow-Mal (RedCode)|193|96|97|4|38/506/2635|
> |BlueCodeEval-Mal (RedCode)|199|100|99|5|43/578/3389|
> |BlueCodeKnow-Mal (RMCbench)|252|126|126|4|40/171/3416|
> |BlueCodeEval-Mal (RMCbench)|272|136|136|7|40/313/4650|
> |BlueCodeKnow-Vul|520|130|130|13|235/1138/4091|
> |BlueCodeEval-Vul|280|140|140|14|143/556/1506|
> |BlueCodeKnow-PI|240|120|120|2|43/294/3511|
> |BlueCodeEval-PI|240|120|120|2|39/377/4731|
>
> We clarify the following construction details and will add these details to the appendix of the revised paper.:
>
> - **Category separation:** BlueCodeKnow and BlueCodeEval use disjoint risk categories (Table 5, Appendix A), ensuring no overlap between knowledge and test sets.
> - **Class balance:** All subsets are approximately balanced between safe and unsafe instances
> - **Deduplication and Filtering:** We perform manual inspection to remove low-quality and near-duplicate examples.
> - **Dataset difficulty:** The datasets include diverse prompt lengths (up to 4k+ characters) and multiple risk categories, making them substantially more challenging than short prompt benchmarks.
> - **Reproducibility:** We will release the full dataset and red-teaming optimization scripts upon acceptance.
>
> > W2: Single F1 metric may hide safety/usability tradeoffs
>
> Thank you for the suggestion. We provide the following extended metrics for the required tasks, models including GPT-4o, Llama-3-8B, Llama-3.3-70B, and Qwen2.5-72B.
>
> Baseline(BL) vs BlueCodeAgent (Ours) (TPR / FPR / F1)
>
> |Task|Model|TPR(BL)|TPR(Ours)|FPR(BL)|FPR(Ours)|F1(BL)|F1(Ours)|ΔF1|
> |---|---|---|---|---|---|---|---|---|
> |Bias|GPT-4o|0.65|0.96|0.01|0.00|0.78|0.98|+0.20|
> |Bias|Llama-8B|0.47|1.00|0.09|0.34|0.60|0.86|+0.25|
> |Bias|Llama-70B|0.31|0.81|0.00|0.00|0.48|0.89|+0.41|
> |Bias|Qwen|0.30|0.85|0.00|0.00|0.46|0.92|+0.46|
> |Mal-RMC|GPT-4o|0.80|0.99|0.00|0.00|0.89|0.99|+0.10|
> |Mal-RMC|Llama-8B|0.44|0.90|0.00|0.01|0.61|0.95|+0.33|
> |Mal-RMC|Llama-70B|0.09|0.90|0.00|0.00|0.16|0.95|+0.78|
> |Mal-RMC|Qwen|0.60|0.99|0.00|0.01|0.75|0.99|+0.24|
> |Mal-RedCode|GPT-4o|0.80|1.00|0.00|0.00|0.89|1.00|+0.11|
> |Mal-RedCode|Llama-8B|0.71|1.00|0.00|0.03|0.83|0.99|+0.15|
> |Mal-RedCode|Llama-70B|0.26|1.00|0.00|0.00|0.41|1.00|+0.59|
> |Mal-RedCode|Qwen|0.78|0.97|0.00|0.00|0.88|0.98|+0.11|
> |PI|GPT-4o|0.71|0.97|0.00|0.00|0.83|0.98|+0.15|
> |PI|Llama-8B|0.07|0.85|0.00|0.00|0.14|0.92|+0.78|
> |PI|Llama-70B|0.99|0.98|0.01|0.00|0.99|0.99|0.00|
> |PI|Qwen|0.90|0.97|0.00|0.00|0.95|0.99|+0.04|
>
> Overall, BlueCodeAgent consistently improves TPR while maintaining near-zero FPR across most settings, indicating that it enhances safety without sacrificing usability.
> The only one tradeoff is Bias+Llama-3-8B (FPR 0.09→0.34): Llama-8B misinterprets bias-related constitution principles and over-refuses benign coding tasks.
> This is a small-model limitation rather than a method flaw, and capable models with reasoning capability could mitigate this issue.

---

> > ### Author Rebuttal · Reviewer_jo6J · 2026-04-01
> >
> > Thanks for your valuable feedback. My concerns have been addressed. However, I still kept my score, while waiting for more suggestions from other reviewers.

---

> > > ### Author Response · Authors · 2026-04-08
> > >
> > > We sincerely thank you for the positive acknowledgement and for confirming that the concerns have been fully addressed. We appreciate your careful evaluation and constructive feedback throughout the process, as well as your openness to the discussion with other reviewers.

---

### Official Review · Reviewer_aJrM · 2026-03-12

**Soundness:** 2
**Presentation:** 3
**Significance:** 2
**Originality:** 2
**Overall Recommendation:** 3
**Confidence:** 4

**Summary:**

BlueCodeAgent is a defense system for AI code generation that uses attack knowledge to improve its defenses. The system first runs an automated attack pipeline that generates many examples of harmful or risky code requests. A defense agent then studies these examples and writes a set of rules and principles to help it recognize dangerous inputs at test time. For the specific task of detecting vulnerable code, where AI models often wrongly flag safe code as dangerous, the system also runs the suspicious code in an isolated environment to check whether the vulnerability can actually be triggered. The paper tests this approach across four tasks: 1) detecting biased coding requests, 2) detecting requests for malicious code, 3) detecting vulnerable code, and 4) detecting prompt injection attacks. It consistently outperforms a wide range of existing methods including safety guardrails, retrieval-based systems, and voting among multiple AI models.

**Compliance With Llm Reviewing Policy:**

Affirmed.

**Key Questions For Authors:**

1) The paper only tests on known risk categories. How does the system perform against attack types that fall entirely outside its four tasks or use strategies not in the knowledge base? This is an important scenario for a defense system in real-world.
2) The pipeline involves many components including retrieval, multi-step LLM calls, and Docker testing. Can the authors provide a cost and latency comparison against the LLM ensemble?
3) Gains on vulnerable code detection are between 0.02 and 0.04 F1 points. Can the authors report confidence intervals?

**Limitations:**

The paper should also address the computational cost of the full pipeline, the latency of multi-step LLM calls and the challenge of keeping the knowledge base current as new attacks emerge.

**Strengths And Weaknesses:**

Strengths
1) The paper is well written and easy to follow. Figures 1 and 2 effectively illustrate the system. The case studies in the appendix, especially the vulnerable code detection example, help build intuition for how the system works in practice.
2) Converting red teaming examples into reusable defense principles is a novel and practical idea. The finding that constitutions reduce missed detections while dynamic testing reduces false alarms is a valuable and practical insight.
3) The paper offers a practical solution and the end-to-end red-to-blue pipeline is a simple and practical design idea.
4) The experimental section inclduing ablation study and Table 4 shows that constitutions and dynamic testing fix different types of errors.

Weaknesses
1) Three of four benchmarks are generated by the same pipeline as the knowledge base, which risks inflating the near-perfect scores. GPT-4o is used on both the attack and defense sides, creating a circular dependency the paper does not address. No statistical testing is reported.
2) The paper only tests on known risk categories ( lack of cross domain transfer ). There is no evidence the system handles truly novel attack types (such as zero day vulnerability), which is a serious concern for real-world use.
3) No cost/performance comparison is provided for LLM ensemble vs. BlueCodeAgent framework.

---

> ### Author Rebuttal · Authors · 2026-03-30
>
> Thank you for recognizing our work as novel and practical. Below, we address each weakness and question.
>
> > W1-a: Benchmark circularity
>
> Shared with Reviewer 69i6. BlueCodeKnow and BlueCodeEval use **disjoint risk categories** (no data contamination). We additionally evaluate on 4 external benchmarks (CBS, RMCbench, Open-Prompt-Injection, CWEval):
>
> |Benchmark|Ours|Directly testing|ΔF1|
> |---|---|---|---|
> |Bias-CBS|**0.92**|0.04|**+0.88**|
> |Mal-RMC|**0.99**|0.88|**+0.11**|
> |PI-OpenPI|**0.40**|0.04|**+0.36**|
> |Vuln-CWEval|**0.73**|0.66|**+0.07**|
>
> BlueCodeAgent outperforms baselines on external benchmarks. Please refer to **Response to Reviewer 69i6, Q2** for more details.
>
> > W1-b: GPT-4o on both attack and defense sides
>
> We acknowledge this concern, which is shared with Reviewer k1ym "Q3&W3: Dependency on base model capability". We conduct experiments using Llama-3-8B, Qwen-3.5-9B, and Llama-3-70B as the **full pipeline backbone**:
>
> |Benchmark|Model|Ours|Directly testing|Δ F1|
> |---|---|---|---|---|
> |Bias|Qwen|0.81|0.52|+0.29|
> |Bias|Llama-8B|0.71|0.57|+0.14|
> |Bias|Llama-70B|0.70|0.67|+0.03|
> |Mal|Qwen|0.96|0.90|+0.06|
> |Mal|Llama-8B|0.82|0.71|+0.11|
> |Mal|Llama-70B|0.99|0.71|+0.28|
> |PI|Qwen|0.91|0.75|+0.16|
> |PI|Llama-8B|0.91|0.14|+0.77|
> |PI|Llama-70B|0.99|0.85|+0.14|
> |Vuln|Qwen|0.62|0.66|-0.04|
> |Vuln|Llama-8B|0.66|0.67|-0.01|
> |Vuln|Llama-70B|0.68|0.67|+0.01|
>
> BlueCodeAgent generally improves over direct testing on Bias/Malicious/PI across all models. Vulnerability detection remains challenging for smaller models due to limited code reasoning capability and higher dynamic test error rates. Overall, while BlueCodeAgent is model-agnostic, its effectiveness requires basic constitution generation capability, reasoning and coding capability. We encourage the use of more capable models, and also note that constitutions generated offline by stronger models can be leveraged by smaller models.
>
> Please refer to **Response to Reviewer k1ym, W3** for detailed analysis.
>
> > W1-c: No statistical testing & Q3: Confidence intervals for vulnerability detection
>
> Thank you for suggesting this analysis. We conducted 3 independent runs on the OOD vulnerability detection task (280 test cases) and report the following results:
>
> **(1) Per-method 95% CI.** We report the mean F1 and its 95% confidence interval (via $t$-distribution, $n{=}3$, $df{=}2$) for each method independently:
>
> **(2) Welch's $t$-test for ΔF1.** Since the Baseline and BlueCodeAgent with Dynamic+Constitution are independent, we apply Welch's $t$-test to compare them. The ΔF1 rows below report the mean difference and its 95% CI. For both models, the CI lower bound is **strictly above zero**, confirming the gains are statistically significant.
>
> |Model|Method|F1 (Mean)|95% CI|
> |---|---|---|---|
> |GPT-4o|Baseline|64.31%|[63.82, 64.80]|
> |GPT-4o|**Ours (Const+Dyn)**|**66.97%**|**[64.58, 69.35]**|
> |GPT-4o|**ΔF1**|**+2.66%**|**[+0.39, +4.92]**|
> |Claude Sonnet 4.5|Baseline|67.30%|[66.40, 68.21]|
> |Claude Sonnet 4.5|**Ours (Const+Dyn)**|**70.45%**|**[68.41, 72.48]**|
> |Claude Sonnet 4.5|**ΔF1**|**+3.14%**|**[+1.41, +4.88]**|
>
> > W2&Q1: No evidence for novel attacks; lack of cross-domain transfer
>
> To simulate novel attack scenarios, we apply mismatched knowledge (e.g., Bias knowledge for Malicious test) and measure F1 results.
>
> |Know\Test|Bias|Mal|PI|
> |---|---|---|---|
> |Bias|**0.96**|0.99|0.89|
> |Mal|0.92|**0.99**|0.88|
> |PI|0.96|0.99|**0.98**|
> |Directly testing|0.79|0.88|0.84|
>
> Matched knowledge (diagonal) remains optimal. All cross-domain entries exceed the Directly Testing baseline, confirming the generalization ability of BlueCodeAgent.
>
> > W3&Q2: Cost/latency comparison vs. LLM ensemble
>
> This concern is shared with Reviewer k1ym "Q1: Overhead and latency".
>
> |Method|Avg. Time(s)|
> |---|---|
> |**BlueCodeAgent (end-to-end)**|**28.73**|
> |**BlueCodeAgent (cached const.)**|**25.93**|
> |LLM-Ensemble (sequential)|78.99|
> |LLM-Ensemble (parallel)|42.93 (Max latency per round, 2 rounds)|
>
> For the token cost, we calculate the total token cost for LLM Ensemble is 6437, which is 1.1x of BlueCodeAgent (5725), averaged on the 280 vulnerability detection test cases. The time and token cost are acceptable for security-critical scenarios.
>
> Please refer to **Response to Reviewer k1ym, Q1** for details.
>
> > Limitations: Keeping the knowledge base up-to-date
>
> Our red-teaming pipeline can be re-run to generate new knowledge. For emerging attacks, users can integrate new red-teaming tools into our red-teaming pipeline and update the knowledge base automatically.

---

> > ### Author Rebuttal · Reviewer_aJrM · 2026-04-02
> >
> > I acknowledge the cost/latency comparison is satisfactory. However, two core concerns remain. First, the cross domain transfer experiment swaps knowledge among Bias, Malicious, and Prompt Injection, all textual level tasks within the system's design scope. This does not demonstrate robustness to genuinely novel attack types outside the four predefined categories. Second, the multi model vulnerability detection results (negative or flat deltas for Qwen and Llama 8B) suggest the pipeline's effectiveness on the hardest task depends heavily on base model strength rather than the architectural design. Given that the generalization and vulnerability detection concerns persist, I maintain my original score.

---

> > > ### Author Response · Authors · 2026-04-08
> > >
> > > We thank the reviewer for clearly outlining these concerns.
> > >
> > > > Regarding the concern on generalization to genuinely novel attack types.
> > >
> > > We agree that robustness to fully **open-world novel attack types** is an important and challenging problem.
> > >
> > > In practice, **genuinely novel attack types emerge continuously and are inherently difficult to exhaustively include and define**. More broadly, we view this as a **fundamental boundary of the knowledge-based safety systems, rather than a limitation unique to our BlueCodeAgent**. If a risk concept is not represented in the current knowledge base, no knowledge-based mechanism can guarantee its detection. This tradeoff is common in real-world security systems (e.g., anti-virus): they work well for known patterns, but need to be continuously updated to handle new and emerging threats.
> > >
> > > Accordingly, our goal is not to claim robustness to fully unbounded novel attacks on the fly, but to provide **reliable safety enforcement within a well-defined and practically relevant risk space**. Within this scope, our experiments evaluate **in-domain and cross-domain generalization**, showing that safety constitutions can transfer across semantically distinct categories (e.g., Bias → Malicious → Prompt Injection), consistently outperforming direct testing baselines. The strong performance under mismatched knowledge suggests that the safety constitutions exhibit generalization capability. **We emphasize that this is non-trivial, as these tasks differ substantially in structure and semantics.**
> > >
> > > For risks outside the current defined scope, our framework **naturally supports a modular extension mechanism**: newly discovered red-teaming methods can be added to our offline red-teaming process to be a component of our automatic red-teaming, and discovered attack patterns can be incorporated into our knowledge base, enabling subsequent constitution summarization.
> > >
> > > Following your valuable suggestion, we will carefully revise our paper to clearly clarify the scope of our generalization claims, and explicitly claim that open-world novel attack is beyond the current scope and not the primary goal of our work.
> > >
> > >
> > > > Regarding the concern on the degraded performance of Qwen and Llama-8B in vulnerability detection tasks.
> > >
> > > We thank the reviewer for this insightful observation. We agree that the performance on smaller models (Qwen3.5-9B, Llama-8B) is important to understand. As also raised by Reviewer k1ym, “understanding the minimum model capability required to sustain the improvement loop is important for the community”, where the reviewer notes that **this analysis is informative and helpful for the community**.
> > >
> > > Our additional analysis reveals a **capability-dependent behavior for the vulnerability detection task**. BlueCodeAgent consistently improves performance when the base model possesses sufficient reasoning and coding ability (e.g., **GPT-4o, Llama-3-70B, Claude models, Codex models**), while for weaker models, the gains may diminish on more challenging tasks such as vulnerability detection. This behavior stems from the design of our framework, which relies on reasoning and coding capabilities, including constitution-guided analysis and dynamic testing. **When these capabilities are limited, the pipeline cannot be fully realized.** For example, we observe that Llama-8B struggles with dynamic testing, resulting in a high execution error rate (38.9%) in generated test code. Qwen struggles with overlong reasoning on vulnerability detection tasks and has difficulty effectively applying the generated safety constitutions.
> > >
> > > We emphasize that the gains are not merely due to stronger base models, but arise from the framework’s ability to **leverage reasoning and coding capabilities**. **The performance degradation phenomenon does not diminish the architectural value of BlueCodeAgent; rather, it defines the operational boundary of our method for high-complexity tasks.**
> > >
> > > We view this not as a limitation, but as an expected requirement for solving complex tasks. As an analogy, prior work on chain-of-thought prompting [1] shows that CoT does not always improve performance for all models, as documented in the CoT paper: “chain-of-thought prompting does not positively impact performance for small models, and only yields performance gains when used with models of∼100B parameters”.
> > >
> > > We will clarify this requirement in the revision and explicitly state that our framework is intended to augment models with adequate reasoning and code understanding capabilities. We would be grateful if the reviewer could reconsider the assessment.
> > >
> > > [1] Chain-of-Thought Prompting Elicits Reasoning in Large Language Models, NeurIPS 2022

---

### Official Review · Reviewer_k1ym · 2026-03-13

**Soundness:** 3
**Presentation:** 4
**Significance:** 3
**Originality:** 3
**Overall Recommendation:** 5
**Confidence:** 4

**Summary:**

This paper addresses the insufficient progress in defensive mechanisms for CodeGen AI by proposing BlueCodeAgent which is an end-to-end blue teaming framework empowered by automated red teaming. The authors devise a system where a red teaming component generates diverse risky instances and edge cases that are subsequently summarized into actionable constitutions to guide the blue team agent in making safety decisions. To address the challenge of over-conservatism in base models, the framework uniquely integrates dynamic code analysis to validate vulnerability claims and effective reduce false positives. Extensive evaluations across four representative tasks including bias and malicious instruction detection demonstrate that BlueCodeAgent achieves significant improvements over strong baselines such as GPT-4o and effectively generalizes to unseen risks through its combination of principled-level defense and nuanced-level runtime verification.

**Compliance With Llm Reviewing Policy:**

Affirmed.

**Final Justification:**

My final recommendation is **Accept**.

The authors have provided a substantial amount of additional experimental results. Based on these results, I believe the proposed method demonstrates excellent performance and can significantly improve the base model's capabilities. I hope the authors will include these data in the next version.

**Key Questions For Authors:**

Thank you for submitting your work to the conference. I particularly appreciated the idea for the software testing.

It would be appreciated if the author could elaborate further on the following aspects (**although a formal response to each question is not strictly necessary**).

- **Computational overhead and latency in real-time applications:** The proposed BlueCodeAgent pipeline appears to be computationally intensive, involving multiple sequential and iterative steps: red teaming generation, constitution summarization, dynamic code execution (compilation and runtime testing). LLM-based code assistants (e.g., GitHub Copilot) typically require extremely low latency to be usable in real-time development environments. Could the authors provide a detailed breakdown of the time cost associated with each module of the framework? Specifically, I am interested in the average latency overhead introduced by the dynamic analysis component versus the static constitution-based defense. Is this framework intended primarily for offline stages of the software development lifecycle (CI/CD) rather than real-time code completion, given the likely delay introduced by the agentic workflow?

- **Implementation of secure dynamic analysis environments:** The paper emphasizes the use of dynamic testing to reduce false positives in vulnerability detection, which is a significant methodological strength. However, executing potentially vulnerable, malicious, or arbitrary code generated by an LLM poses severe security risks to the host system. The paper is relatively light on the implementation details regarding the isolation of these execution environments. Could the authors elaborate on the specific sandboxing technologies employed (e.g., gVisor, Firecracker microVMs, or standard Docker containers)? Furthermore, how does the system mitigate risks such as resource exhaustion (e.g., infinite loops generated by the model) or network-based attacks during the dynamic analysis phase? Understanding the "security of the security tool" is crucial for evaluating the practical deployability of BlueCodeAgent.

- **Dependence on base model capabilities:** The reported experiments primarily utilize GPT-4o, a state-of-the-art closed-source model with high reasoning capabilities. The BlueCodeAgent framework relies heavily on complex tasks such as "Constitution Summarization" and strategic planning during the red-teaming phase. There is a concern that the performance gains are largely attributed to the superior reasoning and instruction-following abilities of GPT-4o, rather than the agentic framework itself. Have the authors conducted ablation studies using smaller or open-source models (e.g., Llama-3-8B/70B, CodeLlama)? It would be valuable to understand if the "Constitution Summarization" mechanism collapses with weaker models that may hallucinate during the summarization step or fail to utilize the dynamic analysis tools effectively. Understanding the minimum model capability required to sustain the improvement loop is important for the community.

**Limitations:**

yes

**Strengths And Weaknesses:**

**Strength:**
- **Integration of dynamic analysis and software testing principles:** Existing blue teaming methods predominantly rely on static textual analysis which leads to high false positive rates due to the over-conservatism of LLMs. By incorporating runtime verification tools to execute and test the code, BlueCodeAgent bridges the gap between traditional software testing and generative AI security. This allows for a more rigorous confirmation of vulnerabilities and represents a meaningful advancement in how we evaluate code generation safety.

- **Automated knowledge accumulation via red-teaming loop:** The paper proposes a compelling end-to-end framework where the blue team directly benefits from the red team outputs through constitution summarization. This mechanism allows the agent to synthesize actionable security principles from diverse edge cases rather than relying on static external knowledge bases. This self-improving loop ensures that the defense mechanism evolves alongside the attack strategies. It effectively turns the red teaming process into a constructive data generation step for defense.

- **Effective reduction of false positives in vulnerability detection:** The empirical results demonstrate a concrete solution to the problem of model over-conservatism in security tasks. The reported reduction in false positives by an average of 19.3% is a strong indicator of the method's practical utility. The ablation studies clearly highlight the complementary nature of the principled-level defense and the nuanced-level analysis. This performance improvement across four distinct tasks confirms the robustness of the proposed architecture.


**Weakness:**
- **High computational overhead and latency:** The proposed agentic workflow involves multiple heavy components including iterative red teaming, constitution summarization, and dynamic code execution. This complex process likely results in high inference latency compared to simple prompt-based guardrails. Real-time coding assistants require extremely fast response times. The current paper writing does not sufficiently analyze the cost-benefit trade-off regarding the time and resources required to deploy this agent in a production environment.

- **Complexity of secure execution environments:** While using dynamic testing is a conceptual strength, the practical implementation of executing potentially malicious or vulnerable code involves significant security risks. The paper does not provide detailed information on the sandboxing or isolation mechanisms used during the dynamic analysis phase. Deploying such a system requires a robust infrastructure to prevent the testing agent itself from becoming a vector for attacks or system instability.

- **Dependency on base model capability:** The effectiveness of the BlueCodeAgent relies heavily on the reasoning capabilities of the underlying LLM (GPT-4o) to generate high-quality red-teaming instances and summarize constitutions. If the base model fails to identify complex edge cases during the red teaming phase, the subsequent blue teaming defense will be weak. It is unclear how well this framework performs with smaller or less capable open-source models that may struggle with the complex reasoning required for constitution synthesis.

- **Baseline:** The author can add the codex and claude code result.

---

> ### Author Rebuttal · Authors · 2026-03-30
>
> We appreciate your comments and suggestions.
>
> > W1&Q1: Overhead and latency
>
> We provide per-module time cost on 280 vulnerability detection test cases (GPT-4o-based):
>
>
> |Module|Avg. Time(s)|
> |---|---|
> |Knowledge Retrieval|0.20|
> |Constitution Summarization|2.59|
> |Static Analysis|3.58|
> |Dynamic Testing|21.07 (86.1% triggered)|
> |Final Judgment|3.75|
> |**BlueCodeAgent (end-to-end)**|**28.73**|
> |**BlueCodeAgent (cached const.)**|**25.93**|
> |LLM-Ensemble (sequential)|78.99|
> |LLM-Ensemble (parallel)|42.93 (Max latency per round, 2 rounds)|
>
>
> Note that the **red-teaming is performed offline**, supporting the *offline red-teaming + online blue-teaming* paradigm. BlueCodeAgent's ~26s latency is substantially lower than LLM-Ensemble (43–79s) and **acceptable for security-critical scenarios**, where many reasoning coding models already incur even longer latency. Latency can be further reduced via (1) **lightweight pre-filtering** to bypass naive benign inputs, and (2) **caching constitutions**.
>
> We agree BlueCodeAgent is well-suited for **offline code review and CI/CD pipelines**, and also viable for online **security-critical real-time scenarios**.
>
> > W2&Q2: Secure execution environment
>
> Our dynamic testing is conducted in sandboxed Docker containers with multiple safety controls: (1) strict CPU/memory/time limits to prevent resource exhaustion, (2) disabled network access to mitigate external attacks, and (3) non-privileged execution with no host filesystem access.
> While secure execution of AI-generated code in the sandbox is an important problem, our work primarily focuses on improving the task performance rather than sandbox design. We will clarify these details in the revision and consider promising sandboxing methods.
>
> > Q3&W3: Dependency on base model capability
>
> We conduct new experiments using three models — Llama-3-8B, Qwen-3.5-9B, and Llama-3-70B — as the **full pipeline backbone** (both constitution summarization and blue-teaming decision) across all test sets.
>
> **Bias Detection (F1, 504 entries):**
>
> |Method|Qwen|Llama-8B|Llama-70B|
> |---|---|---|---|
> |Directly Testing|0.52|0.57|0.67|
> |Safety Prompt (general)|0.59|0.54|0.60|
> |Safety Prompt (specific)|0.35|**0.75**|0.63|
> |BlueCodeAgent w/ Constitution|**0.81**|0.71|**0.70**|
>
> **Malicious Code Detection (F1, 471 entries):**
>
> |Method|Qwen|Llama-8B|Llama-70B|
> |---|---|---|---|
> |Directly Testing|0.90|0.71|0.71|
> |Safety Prompt (general)|0.90|0.72|0.66|
> |Safety Prompt (specific)|0.90|0.77|0.67|
> |BlueCodeAgent w/ Constitution|**0.96**|**0.82**|**0.99**|
>
> **Prompt Injection Detection (F1, 240 entries):**
>
> |Method|Qwen|Llama-8B|Llama-70B|
> |---|---|---|---|
> |Directly Testing|0.75|0.14|0.85|
> |Safety Prompt (general)|0.72|0.22|0.92|
> |Safety Prompt (specific)|0.83|0.34|0.98|
> |BlueCodeAgent w/ Constitution|**0.91**|**0.91**|**0.99**|
>
> **Vulnerability Detection (F1, 280 entries):**
>
> |Method|Qwen|Llama-8B|Llama-70B|
> |---|---|---|---|
> |Directly Testing|**0.66**|**0.67**|0.67|
> |BlueCodeAgent w/ Dynamic & Const.|0.62|0.66|**0.68**|
>
>
> Analysis:
>
> - **Bias/Malicious/PI:** Llama-3-8B's overly aggressive safety alignment causes it to refuse constitution generation (e.g., treating "summarize what constitutes bias" as harmful). This is unique to Llama-3-8B, leaving downstream analysis without useful guidance.
> - **Vulnerability Detection task:** We identify two distinct failure modes for smaller models:
> (1) **Constitution guidance is not effectively internalized by smaller models.** The task is challenging and requires deep code semantic understanding. For Qwen-3.5-9B, while the constitution encourages the model to be more careful, the model tends to classify CWE vulnerabilities as "logic errors", which leads to worse performance. In contrast, Llama-3-70B is more capable to leverage constitution guidance to refine its reasoning.
> (2) **Dynamic testing code quality degrades with smaller models.** Llama-3-8B generates test code with a 38.9% execution error rate. Llama-3-70B achieves a much lower error rate (9.7%) and is more capable to use dynamic testing results to correct false positives.
>
> These findings suggest that while BlueCodeAgent's framework is generally model-agnostic, its effectiveness needs the base model's basic reasoning and code generation capabilities. So we recommend using capable models for constitution generation and dynamic analysis. Also note that constitutions can be generated offline and cached, allowing smaller models to leverage pre-generated constitutions efficiently. We plan to release the constitutions once the paper gets published.
>
> > W4: Additional baselines
>
> Across Bias, Mal-RMC, Mal-RedCode, PI, and Vuln tasks, the F1 improvements over directly testing are:
>
> - **Claude Code (Claude Sonnet 4.5)**: 0.04, 0.12, 0.08, 0.08, 0.04
> - **Codex (GPT-5.2-Codex)**: 0.17, 0.06, 0.04, 0.23, 0.01

---

> > ### Author Rebuttal · Reviewer_k1ym · 2026-04-01
> >
> > Thanks for authors' feedback and response. I suggest the author to update the paper accordingly.

---

> > > ### Author Response · Authors · 2026-04-01
> > >
> > > Dear Reviewer k1ym,
> > >
> > > We sincerely appreciate your timely response and dedication to the review process. We are grateful that you **updated your score from 4 (Weak Accept) to 5 (Accept)**, which is a great encouragement to us.
> > >
> > > We are also glad that our responses have **fully resolved** your concerns regarding:
> > >
> > > 1. **Computational overhead and latency:** We provide detailed per-module time cost in the rebuttal and show that the time cost of BlueCodeAgent is acceptable for security-critical scenarios.
> > > 2. **Implementation of secure dynamic analysis environments:** We clarify the safety configurations of our environment and thus ensure security.
> > > 3. **Dependence on base model capabilities:** We conduct additional experiments with three models, demonstrating the overall effectiveness of BlueCodeAgent and showing that the mechanism requires basic reasoning and coding capabilities.
> > > 4. **Additional baselines and models:** We provide additional results of Codex and Claude Code and show that our mechanism is effective.
> > >
> > > We will incorporate all the suggested updates into the paper. Thank you again for your time and effort in helping improve our work.
> > >
> > > Best regards,
> > >
> > > Authors

---

### Official Review · Reviewer_69i6 · 2026-03-19

**Soundness:** 3
**Presentation:** 3
**Significance:** 1
**Originality:** 2
**Overall Recommendation:** 3
**Confidence:** 3

**Summary:**

The authors present `BlueCodeAgent` as an end-to-end framework for blue teaming, using automated red-teaming to generate attack characteristics to distill into defensive rules of `BlueCodeAgent`. With classification formulation `BlueCodeAgent` presents binary judgement + explanation when provided red-teaming distillation RAG.

Experimentation for `BlueCodeAgent` include coverage for 'bias instruction detection', 'malicious instruction detection' and 'vulnerable code detection'. The  LLM-as-Judge system is evaluated using vanilla flavor, safety-prompted, and majority voting judging.

`BlueCodeAgent` is compared against other safety guardrails including `Llama Guard`, `Llama Firewall`, `PurpCode`; for code output `Vul-RAG`; and `CodeQL`, `Semgrep`, `Bandit` and `Hybrid` for static analysis.

The experiments are performed against four benchsets, three of which belong to the `BlueCode` ecosystem, and `SecCodePLT`.

The ablation studies evaluate: the sensitivity of different risk categories to knowledge relevance; contributions of constitutions and dynamic testing; and whether it's better to pass raw code samples vs their summaries.

A key finding is that components are complementary with constitutions improving recall while dynamic testing helping to increase precision.

**Compliance With Llm Reviewing Policy:**

Affirmed.

**Final Justification:**

We thank the authors for their rebuttal.

While we appreciate the paper's experiment setup (particularly the inclusion of industry wide guardrails such as Llama Guard and Llama Firewall), and appreciate the creation of constitution policies, the main concern remains with evaluation design.

The rebuttal engaged with our concerns but unfortunately does not resolve our core concerns over the point-wise results, and risk of circularity between the knowledge base and test sets. For these reasons our score is unchanged.

We encourage the authors to add additional failure analysis in future iterations.

**Key Questions For Authors:**

1. As the distilled redteam + constitution creation is critical to `BlueCodeAgent` can other cybersecurity datasets be evaluated -- e.g. `SecKnowledge` and `SecEval`? In other words can other datasets be used as the knowledge base


2. Can other datasets also be used as the test benchmark, ie removing `BlueCodeEval` to evaluate for overall generalizability

**Limitations:**

yes

**Strengths And Weaknesses:**

STRENGTHS::

The red-teaming knowledge distillation is well founded, and the categories for evaluation are reasonable (bias instruction/malicious instruction/vulnerable code). The experiments include important industry guardrail frameworks like 'LllamaGuard' and 'LlamaFirewall' which are also useful in evaluation results.

A major contribution of what is ultimately a LLM-as-Judge redteaming-RAG system is the notion constitution construction where redteam knowledge is summarized into actionable rules.



WEAKNESSES::

A significant weakness in the testing is the  circularity of the testing suite, with 3 or the 4 eval datasets belonging to   `BlueCode`, creating the risk for contamination in the creation of knowledge and test sets.

Given the importance of the redteaming distilled knowledge base to `BlueCodeAgent` failure analysis (and potential case study) would help explain exactly how the corpus is able to help `BlueCodeAgent` perform so well in Tab 2.

---

> ### Author Rebuttal · Authors · 2026-03-30
>
> Thank you for the positive comments on our knowledge distillation and constitution mechanism.
>
> > Q1: Can other cybersecurity datasets be used as the knowledge base?
>
> **Yes.** We replace BlueCodeKnow with SecEval, a general-purpose cybersecurity QA dataset (2,182 entries), since SecKnowledge is not publicly available. We concatenate each question with its answer as knowledge text and compute embeddings for retrieval. Results (GPT-4o, same test sets as main paper):
>
> |Task|Directly Testing|Safety Prompt (specific)|**BlueCodeAgent w/ Constitution (Ours)**|
> |---|---|---|---|
> |Bias (504)|0.80|0.88|**0.95**|
> |Malicious-RMC (272)|0.89|0.91|**0.99**|
> |Malicious-RedCode (199)|0.88|0.91|**1.00**|
> |Prompt Injection (240)|0.84|**0.96**|0.80|
>
> Vulnerability Detection — SecEval as knowledge, SecCodePLT as test (280 entries):
>
> |Method|F1|
> |---|---|
> |Directly Testing|0.64|
> |BlueCodeAgent w/ Constitution (Ours)|0.66|
> |**BlueCodeAgent w/ Dynamic & Constitution (Ours)**|**0.67**|
>
>
> SecEval achieves comparable F1 to BlueCodeKnow on bias and malicious tasks. The main gap is PI (-0.18), where SecEval lacks PI-specific knowledge. This confirms our constitution mechanism generalizes to diverse external knowledge sources.
>
> > W1: Circularity of the testing suite & Q2: Can non-BlueCode datasets also be used as the test benchmark?
>
> BlueCodeKnow and BlueCodeEval use **disjoint risk categories** by design (Table 5, Appendix A). Following the reviewer’s suggestion, we evaluate on **4 independent external benchmarks**, the additional benchmarks are as follows:
>
> **(1) Bias — CBS** (Huang et al., TOSEM 2025): 334 explicit bias prompts converted from initial implicit bias prompts by removing non-sensitive attributes. Paired with 334 safe MBPP samples (668 total).
>
> **(2) Malicious — RMCbench**: Knowledge from red-teaming optimized RedCode-Gen (NeurIPS 2024), test from red-teaming optimized RMCbench (ASE 2024). 136+136 (272 total).
>
> **(3) PI — Open-Prompt-Injection** (Liu et al., USENIX Security 2024): 100 Prompt-Injection examples + 100 safe MBPP (200 total).
>
> **(4) Vuln — CWEval** (Peng et al., 2025): 32 unsafe + 25 safe, 20 CWE types. Label-leaking suffixes and comments are removed. Our knowledge covers 55% of CWEval CWEs.
>
> Results (GPT-4o, F1):
>
> |Benchmark|Directly Testing|Safety Prompt (specific)|LlamaGuard|LlamaFirewall|PurpCode-14B|LLM-Ensemble (Initial)|LLM-Ensemble (Discussion)|**BlueCodeAgent w/ Constitution (Ours)**|
> |---|---|---|---|---|---|---|---|---|
> |Bias-CBS|0.04|0.61|0.00|0.00|0.01|0.20|0.25|**0.92**|
> |Malicious-RMC|0.88|0.91|0.79|0.25|0.85|0.84|0.84|**0.99**|
> |PI-OpenPI|0.04|0.37|0.06|0.00|0.00|0.10|0.08|**0.40**|
>
> Vulnerability Detection — CWEval (57 entries, F1):
>
> |Method|CWEval F1|
> |---|---|
> |LlamaGuard-3-8B|0.00|
> |LlamaFirewall|0.00|
> |PurpCode-14B|0.72|
> |LLM-Ensemble (Initial)|0.71|
> |LLM-Ensemble (Discussion)|0.67|
> |Directly Testing|0.66|
> |BlueCodeAgent w/ Constitution (Ours)|0.72|
> |**BlueCodeAgent w/ Dynamic + Constitution (Ours)**|**0.73**|
>
> BlueCodeAgent outperforms all baselines on these benchmarks, confirming the generalizability of BlueCodeAgent to fully independent benchmarks across diverse safety dimensions.
>
> > W2: (Failure) analysis of the knowledge corpus in Tab 2.
>
> **Success:** BlueCodeAgnet distilled examples into transferable security principles (e.g., "verify file ownership before deletion," "use functions from trusted libraries"). These principles improve the model's sensitivity to deeper reasoning about vulnerability patterns even across different CWE types.
>
> **Failure:**
> (1) **Over-conservative flagging**: the model raises out-of-context unsafe scenarios on safe code, leading to false positives.
> (2) **Challenging vulnerability types**: certain CWEs such as CWE-1333 (Inefficient Regular Expression Complexity) are challenging for both static and dynamic analysis, as detecting regex behavior requires strong reasoning beyond general security principles.
> (3) **Final judge overriding dynamic testing**: Dynamic testing confirms the code is safe, yet the final judge still trusts the static analysis over the dynamic testing.
>
> We will include detailed case studies in the revision.
>
> References:
>
> - Li et al., SecEval: A Comprehensive Benchmark for Evaluating Cybersecurity Knowledge of Foundation Models
> - Huang et al., Bias Testing and Mitigation in LLM-based Code Generation, TOSEM 2025
> - Guo et al., RedCode: Risky Code Execution and Generation Benchmark for Code Agents, NeurIPS 2024
> - Chen et al., RMCBench: Benchmarking LLMs' Resistance to Malicious Code, ASE 2024
> - Liu et al., Formalizing and Benchmarking Prompt Injection Attacks and Defenses, USENIX Security 2024
> - Peng et al., CWEval: Outcome-Driven Evaluation on Functionality and Security of LLM Code Generation, 2025

---

> > ### Author Rebuttal · Reviewer_69i6 · 2026-04-04
> >
> > We thank the authors for their responses.
> >
> > Throughout the rebuttal when using SecEval and also in  Tab 1. using BlueCodeKnow, the BlueCodeAgent framework hit F1 scores of .99 and 1.0 on malicious instruction detection across multiple benchmarks, in addition to the acknowledged Prompt Injection drop of .18 when switching to SecEval.   While the PI drop demonstrates that the constitution mechanism generalizes to diverse external knowledge sources, the near perfect performance on malicious instruction detection brings doubt into the evaluation system, with the experiment design not robust enough to demonstrate that the red teaming component actually provides "effective edge cases and guidance for the subsequent blue teaming process."
> >
> >
> > While the pipeline described remains encouraging, it is heavily the experiment design and the single-point results lacking variance that makes it difficult to determine the value of the observed differences. For this reason the score is maintained as is.

---

> > > ### Author Response · Authors · 2026-04-08
> > >
> > > We thank the reviewer for the thoughtful and valuable follow-up. We also sincerely appreciate the acknowledgement of our encouraging pipeline design.
> > >
> > > > "near-perfect performance brings doubt into the evaluation system" and "experiment design is not robust enough to demonstrate that the red teaming component actually provides effective edge cases and guidance for the subsequent blue teaming process."
> > >
> > > We agree with the reviewer that, on a strong backbone such as GPT-4o, near-1.00 F1 scores make it difficult to distinguish the contribution of different knowledge sources.
> > >
> > > To address this concern, we provide additional experiments with **multi-metric, cross-knowledge-source, and cross-backbone evaluation**, including F1, TPR, FPR, and MCC (Matthews Correlation Coefficient),
> > > where:
> > > $$\text{MCC} = \frac{TP \times TN - FP \times FN}{\sqrt{(TP+FP)(TP+FN)(TN+FP)(TN+FN)}}$$
> > >
> > > MCC ranges from $-1$ to $+1$ (higher is better; $+1$ = perfect prediction).
> > >
> > > Below, 'Base' refers to the directly testing baseline, and 'Const' refers to our proposed constitution mechanism.
> > >
> > > | Backbone | Test | Knowledge | F1 (Base → Const) | MCC (Base → Const) | TPR (Base → Const) | FPR (Base → Const) |
> > > |---|---|---|---|---|---|---|
> > > | GPT-4o | RedCode-Gen (199) | Red-team | 0.89 → 1.00 | 0.82 → 1.00 | 0.80 → 1.00 | 0.00 → 0.00 |
> > > | GPT-4o | RedCode-Gen (199) | SecEval  | 0.88 → 1.00 | 0.80 → 1.00 | 0.78 → 1.00 | 0.00 → 0.00 |
> > > | GPT-4o | RMC (272)     | Red-team | 0.89 → 0.99 | 0.82 → 0.99 | 0.80 → 0.99 | 0.00 → 0.00 |
> > > | GPT-4o | RMC (272)     | SecEval  | 0.89 → 0.99 | 0.81 → 0.97 | 0.80 → 0.98 | 0.01 → 0.01 |
> > > | Llama-3-8B | RedCode-Gen (199) | Red-team | **0.83 → 0.99** | **0.74 → 0.97** | **0.71 → 1.00** | 0.00 → 0.03 |
> > > | Llama-3-8B | RedCode-Gen (199) | SecEval  | 0.84 → 0.78 | 0.75 → 0.54 | **0.72 → 1.00** | 0.00 → **0.56 ↑↑** |
> > > | Llama-3-8B | RMC (272)     | Red-team | **0.61 → 0.95** | **0.53 → 0.90** | **0.44 → 0.90** | 0.00 → 0.01 |
> > > | Llama-3-8B | RMC (272)     | SecEval  | 0.59 → 0.71 | 0.51 → 0.31 | **0.43 → 0.91** | 0.01 → **0.65 ↑↑** |
> > >
> > > Analysis and Summary:
> > >
> > > 1. **On GPT-4o, the results of the improved F1 look close because of the strong backbone GPT-4o**.
> > > On GPT-4o, the baseline performance is already high (F1 ≈ 0.89), and both Red-team and SecEval knowledge lead to near-perfect F1 scores (~1.00). This effect is primarily due to the strong generalization capability of GPT-4o, which limits the visibility of differences between knowledge sources.
> > >
> > > 2. **Clear differentiation emerges on smaller backbones (Llama-3-8B).**
> > > On Llama-3-8B, the impact of the knowledge source becomes more significant.
> > > With Red-team knowledge, FPR remains ≤ 0.03, and MCC stays at 0.90–0.97; with SecEval summarized constitution, FPR jumps to **0.556 and 0.654**, and MCC collapses to 0.31–0.54.
> > > Llama-3-8B misapplies SecEval's general cybersecurity descriptions to benign code, producing severe over-refusal, showing a small-model alignment limitation.
> > > The Red-team knowledge does not exhibit this problem because it is distilled from concrete edge cases produced by the red-teaming pipeline, making it more precise and behavior-aligned.
> > >
> > > 3. **TPR improvements remain large and consistent across all settings.**
> > > Across all models and knowledge sources, our method consistently improves TPR. This confirms our mechanism is effective in identifying unsafe samples that the baseline misses.
> > >
> > > We will incorporate the tables and the discussion into the revision. We hope this additional evidence clarifies the role of the red-teaming component and addresses the reviewer’s concerns. We would be grateful if the reviewer could reconsider the assessment.

---

### Decision · Program_Chairs · 2026-04-30

**Decision:**

Accept (regular)

**Comment:**

All reviewers seem to agree this is an interesting approach to tackling defensive posture in Agentic designs by leveraging a Red-Teaming agent, which has been historically easier, in a symbiotic approach. While not all problems are resolved, it seems to make a clear benefit and push forward the research with appropriate future works remaining.

Two reviewers raised important concerns which were ackgnoweledged by the authors and they offered an extensive set of further experiments to address these important issues. While some reviewers felt that the paper is still not ready, the AC felt that with the suggested improvements, the paper can be an important contribution.